# Dopaminergic Signalling Enhances IL-2 Production and Strengthens Anti-Tumour Response Exerted by Cytotoxic T Lymphocytes in a Melanoma Mouse Model

**DOI:** 10.3390/cells11223536

**Published:** 2022-11-09

**Authors:** Ornella Chovar-Vera, Ernesto López, Felipe Gálvez-Cancino, Carolina Prado, Dafne Franz, Diego A. Figueroa, Alexandra Espinoza, Claudio Figueroa, Alvaro Lladser, Rodrigo Pacheco

**Affiliations:** 1Laboratorio de Inmunoncología, Fundación Ciencia & Vida, Ñuñoa, Santiago 7780272, Chile; 2Laboratorio de Neuroinmunología, Fundación Ciencia & Vida, Ñuñoa, Santiago 7780272, Chile; 3Centro Cientifico y Tecnologico de Excelencia Ciencia & Vida, Fundacion Ciencia & Vida, Ñuñoa, Santiago 7780272, Chile; 4Facultad de Medicina y Ciencia, Universidad San Sebastián, Providencia, Santiago 7510156, Chile

**Keywords:** CD8^+^ T-cells, dopamine, tumour

## Abstract

Dopamine has emerged as an important regulator of immunity. Recent evidence has shown that signalling through low-affinity dopamine receptors exerts anti-inflammatory effects, whilst stimulation of high-affinity dopamine receptors potentiates immunity in different models. However, the dopaminergic regulation of CD8^+^ T-cells in anti-tumour immunity remains poorly explored. Here, we studied the role of dopamine receptor D3 (DRD3), which displays the highest affinity for dopamine, in the function of CD8^+^ T-cells and its consequences in the anti-tumour immune response. We observed that the deficiency of *Drd3* (the gene encoding DRD3) in CD8^+^ T-cells limits their in vivo expansion, leading to an impaired anti-tumour response in a mouse melanoma model. Mechanistic analyses suggest that DRD3 stimulation favours the production of interleukin 2 (IL-2) and the surface expression of CD25, the α-chain IL-2 receptor, which are required for expansion and effector differentiation of CD8^+^ T-cells. Thus, our results provide genetic and pharmacologic evidence indicating that DRD3 favours the production of IL-2 by CD8^+^ T-cells, which is associated with higher expansion and acquisition of effector function of these cells, promoting a more potent anti-tumour response in a melanoma mouse model. These findings contribute to understanding how dopaminergic signalling affects the cellular immune response and represent an opportunity to improve melanoma therapy.

## 1. Introduction

Several studies have shown dopamine as a major regulator of immune homeostasis. Consequently, the dysregulation of dopamine levels or the expression of dopamine receptors has been consistently involved in the physiopathology of autoimmunity and neurological disorders that are associated with neuroimmune regulation [1,2]. Accordingly, several subsets of immune cells express dopamine receptors, and their immune activity, migratory pattern, differentiation, and proliferation are affected by dopaminergic stimulation [3]. The precise effect of dopaminergic stimulation on immune homeostasis depends on the particular subset of immune cells, the dopamine receptors expressed by those cells, and the dopamine levels available in the particular tissue. In this regard, signalling through high-affinity dopamine receptors, including dopamine receptors D3 (DRD3), DRD4, and DRD5 [4], has been consistently involved in promoting an inflammatory behaviour in immune cells, whereas signalling through low-affinity dopamine receptors, especially DRD1 and DRD2, has been associated with anti-inflammatory downstream effects [5,6].

Dopaminergic signalling through DRD3 is especially interesting in immune cells, as this receptor displays the highest affinity for dopamine (Ki ≈ 27 nM) [7]. Accordingly, this receptor is selectively stimulated in the presence of very low levels of dopamine. This means that DRD3 acts as a danger sensor when it is selectively stimulated in organs that generally contain high levels of dopamine (in the micromolar order), such as the colonic mucosa or the striatum, among others [3]. Remarkably, these reductions in dopamine levels have been associated with tissue inflammation [8,9]. Thus, reduction in dopamine levels has been shown to selectively stimulate DRD3-signalling in regulatory T-cells (Treg) [10], effector CD4^+^ T-cells [11,12,13], B-cells [14], macrophages [15], and glial cells [16], promoting inflammation. For instance, a group of articles investigating the role of DRD3 on CD4^+^ T-cells upon colonic inflammation or neuroinflammation has described that DRD3-signalling on these cells potentiates the generation of the T helper 1 (Th1) subset of effector cells, whilst favouring the expansion of Th17 cells [11,13,17]. Furthermore, DRD3-signalling on Treg has been shown to change their migratory pattern and tissue tropism and also reduces their production of IL-10, thus attenuating their suppressive activity in the inflamed gut mucosa [10]. In addition, a recent study showed that DRD3-signalling on B-cells regulates CXCR3 expression and their infiltration into the central nervous system (CNS) upon neuroinflammation [14]. Moreover, pharmacologic evidence has suggested that DRD3-signalling in macrophages induces the acquisition of the pro-inflammatory M1 profile [15].

The selective stimulation of DRD3 is not only given upon the reduction of dopamine levels in tissues that normally contain high levels of dopamine (like the striatum or the colonic mucosa) but also in some organs where dopamine sometimes appears and is at low levels, such as lymphoid organs. In this regard, primary and secondary lymphoid organs receive sympathetic inputs that release dopamine at the nanomolar range [3]. The release of dopamine in secondary lymphoid organs, such as the spleen and lymph nodes, is of particular interest, as in these organs antigen-presenting cells (APCs) activate T-cells, thus triggering the development of the adaptive immune response [18]. In this scenario, nanomolar levels of dopamine may stimulate high-affinity dopamine receptors expressed on T-cells and APCs, shaping the immune response.

Although several studies have addressed the role of the dopaminergic regulation of the immune response in autoimmunity, neurodegenerative disorders, and some neuropsychiatric diseases [1,2,19], the role of dopamine in anti-tumour immunity has been poorly explored. CD8^+^ T-cells play a central role in the anti-tumour immune response. After recognising tumour antigens presented by dendritic cells (DCs; which are professional APCs) in the secondary lymphoid organs, CD8^+^ T-cells become activated and differentiate into cytotoxic T lymphocytes (CTL), which may directly enact the killing of tumour cells in response to the recognition of tumour antigens [20]. CTL kill target cells by releasing the cytotoxic molecules perforin and granzyme B, as well as effector cytokines, such as interferon γ (IFN-γ) and tumour necrosis factor α (TNF-α). In addition, CTL may promote the anti-tumour activity of macrophages and natural killer (NK) cells [21]. B-cells might also contribute to the anti-tumour response by producing antibodies (Abs) directed to surface tumour antigens, thereby coupling the tumour recognition with the inflammatory effector response associated to the Fc fragment of Abs [22].

Notably, most of the relevant actors involved in the anti-tumour immune response have been described to express DRD3, including CD8^+^ T-cells, effector CD4^+^ T-cells, Treg, macrophages, NK cells, B-cells, and DCs [10,13,23,24,25]. Only the role of DRD3-signalling in DCs has been addressed in the context of the anti-tumour response. In this regard, it has been shown that DRD3-signalling on DCs reduces the antigen cross-presentation to CD8^+^ T-cells, dampening the anti-tumour response [24]. Here, we have addressed the role of DRD3-signalling on the leading actor of anti-tumour response, the CD8^+^ T-cells. For this purpose, we used a mouse model of melanoma and tumour-specific CD8^+^ T-cells. Our findings provide evidence indicating that a genetic deficiency of DRD3 impairs the production of IL-2 by CD8^+^ T-cells, which is associated with attenuated expansion and acquisition of effector function by these cells, and with reduced efficacy of the anti-tumour response in a melanoma mouse model.

## 2. Materials and Methods

### 2.1. Animals

Wild-type (*Drd3^+/+^*; *Cd45.2*^+/+^) and OT-I mice were obtained from The Jackson Laboratory (Bar Harbor, ME, USA). Of note, OT-I mice (strain 003831 from The Jackson Laboratory), which harbour chicken ovalbumin (OVA)-specific CD8^+^ T-cells expressing a transgenic T-cell receptor (TCR) bearing the variable chains Vα2 and Vβ5 with specificity for the MHC/peptide complex H-2K^b^/OVA_257-264_ (see Appendix A), were not developed in the *Rag1^−/−^* background. *Drd3^−/−^* mice were kindly donated by Dr. Marc Caron, which were initially generated in the 129SvJ strain [26], and then backcrossed in the C57BL/6 genetic background for more than ten generations. Wild-type mice (*Drd3^+/+^*; *Cd45.1*^+/+^) were kindly provided by Dr. María Rosa Bono. *Cd45.2*^+/+^
*Drd3^−/−^* OT-I mice and *Cd45.1*^+/−^
*Cd45.2*^+/−^
*Drd3^+/+^* OT-I mice were generated by crossing parental mouse strains. We confirmed these new strains to be transgenic and *Drd3*-deficient by flow cytometry analysis of blood cells and PCR of genomic DNA, respectively. All mouse strains were from the C57BL/6 genetic background. Mice from 6 to 8 weeks of age were used in all experiments. All studies were carried out according to the 8th edition of the Guide for the Care and Use of Laboratory Animals. Experimental protocols were approved by the IACUC at the Fundación Ciencia & Vida, including those involving anaesthesia, pain, distress, and euthanasia (Permit number P-024/2021). The ethical approval date is 12 May 2021.

### 2.2. Reagents

The following reagents were purchased from BioLegend (San Diego, CA, USA): Low endotoxin Azide-free (LEAF) purified anti-mouse CD3ε Ab (clone 195-2C11) and anti-mouse CD28 Ab (clone 27.51); FITC-, PE- and allophycocyanin- coupled anti-CD3 Ab (clone 17A2); Brilliant Violet 421- and allophycocyanin-Cy7- coupled anti-CD8 Ab (clone 53-6.7); Brilliant Violet 421- and PE- coupled anti-Vα2 Ab (clone B20.1); allophycocyanin-coupled anti-Vβ5 Ab (clone MR9-4); FITC- and PE-Cy7-coupled anti-CD45.1 Ab (clone A20); PerCP/Cy5.5-coupled anti-CD45.2 Ab (clone 104); PE- and PerCP-coupled anti-CD44 Ab (clone IM7); allophycocyanin-coupled anti-CD62L Ab (clone MEL-14); FITC-coupled anti-CD25 Ab (clone 3C7); PE-coupled anti-IFN-γ Ab (clone XMG1.2); PE/Cy7-coupled anti-IL-2 Ab (clone JES6-5H-4); recombinant mouse IL-2 (carrier-free) (cat # 575406); and Zombie Aqua (ZAq) fixable viability kit (cat # 423101). Phorbol 12-myristate 13-acetate (PMA; cat # 16561-29-8), ionomycin (cat # I0634), and brefeldin A (cat # B6542) were obtained from Sigma-Aldrich (San Louis, MO, USA). Roswell Park Memorial Institute (RPMI) medium (cat # A1049-01), fetal bovine serum, L-glutamine, penicillin and streptomycin were purchased from Life Technologies (Carlsbad, CA, USA). Phosphate buffer saline (PBS) was obtained from Thermo Fisher Scientific (cat # 21-040-CVR).

### 2.3. CD8^+^ T-Cell Isolation and Activation In Vitro

Splenic CD8^+^ T-cells were purified from secondary lymphoid organs of *Drd3^+/+^* or *Drd3^−/−^* OT-I mice using the EasySep Mouse CD8^+^ T-cell Enrichment Kit (StemCell Technologies; cat # 19853). All in vitro experiments were performed using a complete RPMI medium (supplemented with 10% FBS, 2 mM L-glutamine, 100 U/mL penicillin, and 100 μg/mL streptomycin). CD8^+^ T-cells (5 × 10^5^ cells/well) were activated with 1 μg/mL of plate-bound anti-CD3 and anti-CD28 Abs on flat bottom 24-well plates (Thermo Scientific) for 24 h, and the culture supernatant was collected for IL-2 quantification by ELISA as previously described [27]. In other experiments, CD8^+^ T-cells were initially activated with plate-bound anti-CD3 and anti-CD28 Abs, with or without recombinant IL-2 (50 U/mL), for 48 h. Afterwards, cells were washed with fresh medium and then cultured on flat bottom 24-well plates in the absence of anti-CD3 and anti-CD28 Abs for an additional 48 h. Finally, cells were stimulated or not with 1 μg/mL of OVA_257-264_ peptide (pOT-I) for 6 h. Brefeldin A (1 μg/mL) was added during the last 4 h.

### 2.4. Determination of ERK1/2 Phosphorylation

ERK1/2 phosphorylation was evaluated as previously described [17]. Briefly, CD8^+^ T-cells were isolated (see Section 2.3), washed twice, and resuspended in pre-warmed RPMI medium and incubated in the absence or presence of either 1 μg mL^−1^ anti-CD3ε Ab (Biolegend, San Diego, CA, USA) or 50 nM PD128907 (Tocris, Bristol, UK) for 5 min. Cells were treated with lysis buffer (0.5% SDS, 12.5 mM Tris-HCl pH 6.8, 2.5% glycerol, 0.0025% bromophenol blue and protease and phosphatase inhibitor cocktail (both from Roche, Mannheim, Germany)). Protein concentrations were estimated using the bicinchoninic acid (BCA) method (Thermo Scientific, Rockford, IL, USA). Cell lysates (50 μg sample^−1^) were resolved by SDS-PAGE, transferred to polyvinylidene difluoride membranes (PVDF, Thermo Scientific, Rockford, IL, USA), and diphosphorylated-ERK1/2 were detected using a mouse phospho-specific ERK1/2 mAb (1:10,000; Sigma-Aldrich, St. Louis, MO, USA) followed by HRP conjugated goat anti-mouse IgG Ab (1:5000; Rockland, Gilbertsville, PA, USA). Immunodetection was conducted with SuperSignal West Pico chemiluminescent substrate (Thermo Scientific, Rockford, IL, USA). Membranes were stripped and reprobed with rabbit anti-ERK1/2-specific pAb (1:10,000; Sigma-Aldrich, St. Louis, MO, USA) followed by HRP conjugated goat anti-rabbit Ab (1:5000; Rockland, Gilbertsville, PA, USA) and detected as described above.

### 2.5. Flow Cytometry Analysis

Cells were first incubated with a ZAq fixable viability kit and immunostained with fluorochrome-coupled Abs for cell surface markers at 4 °C and in darkness for 30 min. When only cell surface markers were analysed, cells were washed twice with 2% FBS in PBS and resuspended in a final volume of 200 μL. Alternatively, cells were permeabilized and fixed for intracellular cytokine staining using a Cytofix/Cytoperm Fixation/Permeabilization solution (BD Bioscience, cat # 554714) according to the manufacturer’s instructions. Cells were immunostained with fluorochrome coupled Abs for cytokines at 4 °C and in darkness for 30 min, and then washed twice with 2% FBS in PBS and resuspended in a final volume of 200 μL. Data were collected with a FACSCanto II (BD Bioscience, San Jose, CA, USA), and results were analysed with FACSDiva (BD Bioscience, San Jose, CA, USA) and FlowJo software (Tree Star, Ashland, OR, USA; accessed on 19 August 2021 new v9/10; https://www.flowjo.com).

### 2.6. Vaccination

Mice were intradermally vaccinated in the lower back skin with 40 µg of the pVAX-OVA plasmid, which encodes a membrane-bound form of the chicken OVA. Immediately, DNA electroporation was performed by placing a parallel needle array electrode (two rows of four 2 mm pins, 1.5 × 4 mm gaps; 47-0040, BTX electroporation system) on the injected skin area. Two electric pulses of 1125 V/cm (0.05 ms each) followed by eight electric pulses of 275 V/cm (10 ms each) were given using the AgilePulse In Vivo System (BTX Molecular Delivery System, ref 47-0400N), as described in [28]. Plasmids were purified using a NucleoBond Xtra Midi (cat # 740420.10) according to the manufacturer’s instructions.

### 2.7. Tumour Challenge

Thirteen days after vaccination, mice were subcutaneously (s.c.) inoculated in the lower back skin, close to the vaccination site, with 50 µL of PBS containing 10^6^ tumour cells (B16/pOT-I melanoma cells, which were generated by transduction of B16F10 cells with a lentiviral vector encoding the pOT-I epitope in a minigene [29]). Survival and tumour growth were monitored in the mice over time. To quantify tumour size, the diameter of the tumour was measured perpendicularly using a calliper. Tumour volume was calculated using the following formula: V = (D × d^2^)/2 where V is the volume (mm^3^), D is the larger diameter (mm), and d is the smaller diameter (mm) as previously described [30,31]. According to the approved ethical protocol, mice were killed when moribund or when the mean tumour diameter was ≥15 mm.

### 2.8. Statistical Analysis

Statistical analysis was performed using a two-tailed unpaired Student’s *t*-test when comparing only two groups and with one-way ANOVA followed by Tukey’s post hoc test when comparing more than two groups with only one variable (treatment or genotype). Differences between groups in survival were analysed by the logrank test (Mantel–Cox). Two-way ANOVA followed by Tukey’s post hoc test was performed to analyse differences in experiments comparing different genotypes and different treatments. All analyses were carried out using GraphPad Prism 9 Software (accessed on 25 March 2022 version 9.4.0.1; https://www.graphpad.com/scientific-software/prism/). *p* values < 0.05 were considered significant.

## 3. Results

### 3.1. Drd3-Deficient CD8^+^ T-cells Exhibit Impaired Antigen-Induced Expansion and Reduced IFN-γ Production In Vivo

To address the role of DRD3 in the activation and acquisition of the effector function of CD8^+^ T-cells, we used an antigen-specific approach in vivo. To this end, wild-type mice received an i.v. transfer of *Drd3^+/+^* or *Drd3^−/−^* OT-I transgenic CD8^+^ T-cells, which express a T-cell receptor (TCR) specific for the recognition of the chicken ovalbumin (OVA)-derived peptide OVA_257-264_ (pOT-I) over the H-2K^b^ molecule [32]. A control group of mice did not receive OT-I transgenic CD8^+^ T-cells. One day later, recipient mice were intradermally immunized with a DNA vaccine encoding OVA (pVAX-OVA). To analyse how DRD3 signalling affects the potency of CD8^+^ T-cell activation and their effector function, eleven days after the vaccination, the extent of OT-I CD8^+^ T-cells’ expansion was determined in peripheral blood by staining the transgenic TCR, which is composed of Vα2 and Vβ5 chains (Appendix A), and ex vivo peptide stimulation followed by intracellular IFN-γ staining. OVA immunization induced a substantial expansion of Vα2^+^ Vβ5^+^ CD8^+^ T-cells in the mice receiving *Drd3*-sufficient OT-I CD8^+^ T-cells compared with the mice that did not receive exogenous lymphocytes (Figure 1A). Strikingly, mice receiving *Drd3*-deficient OT-I CD8^+^ T-cells displayed a marked impairment in the expansion of Vα2^+^ Vβ5^+^ CD8^+^ T-cells, compared to mice receiving *Drd3*-sufficient OT-I CD8^+^ T-cells (Figure 1A). Indeed, the expansion of Vα2^+^ Vβ5^+^ CD8^+^ T-cells in mice harbouring *Drd3*-deficient OT-I was similar to the expansion of endogenous Vα2^+^ Vβ5^+^ CD8^+^ T-cells in control mice that were not immunized (Figure 1A). Similar results were obtained in males and females (Appendix A). Furthermore, to evaluate the frequency of antigen-specific CD8^+^ T-cells and how DRD3 signalling impacts the effector function, we determined the levels of IFN-γ production by CD8^+^ T-cells following ex vivo stimulation with OT-I peptide. The results show that *Drd3*-deficiency in CD8^+^ T-cells resulted in a significant reduction in the percentage of cells producing IFN-γ in response to pOT-I (Figure 1B). Nevertheless, the intensity of IFN-γ production (determined as the mean fluorescence intensity; MFI) observed for *Drd3*-sufficient and *Drd3*-deficient CD8^+^ T-cells in response to pOT-I was similar (Figure 1B). Of note, the effect of *Drd3*-deficiency in CD8^+^ T-cells on the intensity and the percentage of IFN-γ production in response to pOT-I was similar in males and females (Appendix A). Even though the expansion of *Drd3*-deficient OT-I CD8^+^ T-cells was similar to the extent of expansion of endogenous Vα2^+^ Vβ5^+^ CD8^+^ T-cells present in control mice (Figure 1A), the group of *Drd3*-deficient OT-I CD8^+^ T-cells contained a higher percentage of cells producing IFN-γ in response to pOT-I, in comparison to the control group (Figure 1B). This discrepancy might be because the extent of expansion was analysed in the Vα2^+^ Vβ5^+^ CD8^+^ gate (Figure 1A), whilst the IFN-γ production was determined in the total CD8^+^ population in response to the ex vivo restimulation with pOT-I (Figure 1B). Therefore, it is possible that some cells producing IFN-γ detected in the *Drd3*-deficient group correspond to endogenous pOT-I specific CD8^+^ T-cells. Although this fact may have led to an overvalue in the percentage of cells producing IFN-γ in the *Drd3*-deficient group, this does not change the main conclusion of these experiments, which indicates that the percentage of cells producing IFN-γ is lower in *Drd3*-deficient OT-I CD8^+^ T-cells, compared with *Drd3*-sufficient OT-I CD8^+^ T-cells. Together, these results indicate that DRD3 signalling on CD8^+^ T-cells strengthens the expansion and the acquisition of the effector function induced by TCR stimulation.

### 3.2. Genetic Drd3-Deficiency Does Not Affect the Migratory Pattern of CD8^+^ T-Cells through Secondary Lymphoid Organs

A previous study provided pharmacological evidence indicating that DRD3 stimulation of CD8^+^ T-cells potentiates chemotaxis towards CXCL19 and CXCL21, the CCR7 ligands that lead the circulation of these cells through secondary lymphoid organs [25]. According to those findings, we theorised that the impaired expansion observed in mice harbouring *Drd3*-deficient CD8^+^ T-cells could be due, at least in part, to a reduced migration of these cells through the secondary lymphoid organs, where APC would present the tumour antigens. For this purpose, we i.v. transferred CD8^+^ T-cells from congenic *Drd3^+/+^* (*Cd45.1^+/−^ CD45.2^+/−^*) or *Drd3^−/−^* (*Cd45.1^−/−^ CD45.2^+/+^*) OT-I mice mixed in a 1:1 ratio into wild-type (*Cd45.1^+/+^ CD45.2^−/−^*) recipient mice. One day later, the proportion of *Drd3^+/+^* and *Drd3^−/−^* CD8^+^ T-cells infiltrating the inguinal lymph nodes (ILN), mesenteric lymph nodes (MLN), and the spleen was analysed by flow cytometry using the congenic surface markers. The results show that *Drd3*-deficiency in CD8^+^ T-cells does not affect the extent of infiltration of these cells in any of the secondary lymphoid organs analysed (Figure 2A–C). To analyse in more detail whether DRD3 signalling exerts an effect on the migratory pattern of CD8^+^ T-cells through secondary lymphoid organs, we also determined the expression of the chemokine receptor CCR7 and the cell adhesion molecule CD62L on the cell surface of *Drd3^+/+^* and *Drd3^−/−^* CD8^+^ T-cells in the spleen, ILN, and MLN. The results show that *Drd3*-deficiency does not affect the expression of either CCR7 or CD62L (Figure 2D,E). Together, these findings provide genetic evidence suggesting that DRD3signalling does not affect the migratory programme of naïve CD8^+^ T-cells through the secondary lymphoid organs.

### 3.3. DRD3 Signalling Reinforces IFN-γ Production by Enhancing IL-2 Secretion

TCR stimulation on T-cells induces IL-2 secretion, which constitutes an autocrine growth factor for T-cells, promoting their proliferation and favouring their subsequent differentiation into effector T-cells [33,34]. Since our results indicated that *Drd3*-deficiency impaired the expansion and acquisition of effector function induced by TCR stimulation on CD8^+^ T-cells, we hypothesised that DRD3 stimulation might favour IL-2 production. To address this hypothesis, we compared IL-2 production by *Drd3^+/+^* and *Drd3^−/−^* OT-I CD8^+^ T-cells in response to in vitro antigen stimulation. In accordance with our hypothesis, the results show that *Drd3^−/−^* OT-I CD8^+^ T-cells showed an impaired IL-2 production upon OT-I peptide stimulation (Figure 3A). In addition, we confirmed that *Drd3* deficiency results in reduced secretion of IL-2 into the culture supernatant upon TCR stimulation (Figure 3B). We next sought to address whether the reduced IFN-γ production observed in *Drd3*-deficient CD8^+^ T-cells (Figure 1B) is a consequence of the decreased IL-2 production observed in these cells. To this end, we activated *Drd3^+/+^* and *Drd3^−/−^* OT-I CD8^+^ T-cells in the presence or absence of exogenous IL-2, and the extent of IFN-γ production was determined by flow cytometry. The results show that *Drd3^−/−^* OT-I CD8^+^ T-cells produced a significantly lower degree of IFN-γ than *Drd3^+/+^* OT-I CD8^+^ T-cells in response to pOT-I in the absence of exogenous IL-2. Nevertheless, this difference in IFN-γ observed between both genotypes was abrogated in the presence of exogenous IL-2 (Figure 3C). Thus, these results suggest that the reduced IFN-γ production observed on *Drd3*-deficient CD8^+^ T-cells is a consequence of the impaired production of IL-2 by these cells. A critical step in the action mechanism of IL-2 during T-cell expansion is the autocrine upregulation of the α-chain of the IL-2 receptor (CD25), which increases the affinity of the IL-2 receptor about 100 fold, making T-cells substantially more sensitive to IL-2 [35]. Accordingly, we performed further experiments to determine how CD25 expression was affected by DRD3 signalling on CD8^+^ T-cells. Similar to the results obtained for IFN-γ production, the surface CD25 expression was significantly lower on OT-I peptide-stimulated *Drd3*-deficient CD8^+^ T-cells, compared to *Drd3*-sufficient CD8^+^ T-cells in the absence of IL-2, whereas reduced CD25 expression was abolished upon treatment with exogenous IL-2 (Figure 3D). We next sought to evaluate whether the pharmacologic stimulation of DRD3 favours the production of IL-2. ERK1/2 is one of the signalling pathways triggered by the TCR activation that has been shown to be modulated by DRD3 stimulation [17]. Therefore, we evaluated how a selective DRD3 agonist (PD128907) affected the phosphorylation of ERK1/2 in TCR-stimulated *Drd3^+/+^* and *Drd3^−/−^* CD8^+^ T-cells. The results show that DRD3 stimulation reduced the extent of ERK1/2 phosphorylation induced by the TCR activation in *Drd3*-sufficient CD8^+^ T-cells, an effect that was abrogated in *Drd3*-deficient CD8^+^ T-cells (Figure 4A). Of note, in the absence of TCR activation, the DRD3 stimulation did not affect ERK1/2 phosphorylation in either *Drd3^+/+^* or *Drd3^−/−^* CD8^+^ T-cells (Figure 4A). Afterwards, we confirmed that, in the absence of exogenous IL-2, the treatment of *Drd3^+/+^* OT-I CD8^+^ T-cells with the DRD3 agonist exacerbated the production of IL-2 and the surface expression of CD25 (Figure 4B,C). Of note, the differences in IL-2 production and CD25 expression observed between DRD3 stimulated and untreated *Drd3^+/+^* OT-I CD8^+^ T-cells were abolished in the presence of exogenous IL-2 (Figure 4B,C). Together, these results provide genetic and pharmacological evidence suggesting that DRD3 signalling strengthens the expression of IL-2 and its high-affinity receptor and consequently favours the acquisition of the effector function by CD8^+^ T-cells.

### 3.4. Drd3-Deficiency in CD8^+^ T-Cells Results in Reduced Potency in the Anti-Tumour Response

To test the relevance of DRD3 signalling on CD8^+^ T-cells in the anti-tumour response, a mouse model of melanoma was induced in mice bearing *Drd3*-sufficient or *Drd3*-deficient tumour-specific CD8^+^ T-cells. For this purpose, wild-type C57BL/6 mice received the i.v. transfer of *Drd3^+/+^* or *Drd3^−/−^* OT-I transgenic CD8^+^ T-cells. A control group of mice did not receive OT-I transgenic CD8^+^ T-cells. One day later, experimental mice were vaccinated with pVAX-OVA. Two weeks later, experimental mice were inoculated with B16 melanoma cells expressing the pOT-I (B16/pOT-I cells [29]), and tumour growth and mice survival were monitored periodically (see the experimental design in Appendix A and Figure 5A). The results show that mice receiving *Drd3*-sufficient or *Drd3*-deficient OT-I transgenic CD8^+^ T-cells displayed a significant reduction in tumour growth over time, compared with those mice that did not receive OT-I transgenic CD8^+^ T-cells (Figure 5B,C). Furthermore, mice receiving *Drd3^−/−^* OT-I transgenic CD8^+^ T-cells showed faster tumour growth, compared with those receiving *Drd3^+/+^* OT-I transgenic CD8^+^ T-cells (Figure 5B,C). Similar results were obtained in males (Figure 5B,C) and females (Appendix A). According to this observation, male mice bearing *Drd3*-sufficient OT-I CD8^+^ T-cells displayed a significantly higher survival rate over time, in comparison with those bearing *Drd3*-deficient OT-I CD8^+^ T-cells (Figure 5D). Although the results obtained with female mice displayed similar survival curves, there were no significant differences between the groups of mice receiving *Drd3*-sufficient and *Drd3*-deficient OT-I CD8^+^ T-cells (Appendix A). Moreover, male or female mice receiving *Drd3*-sufficient or *Drd3*-deficient OT-I transgenic CD8^+^ T-cells showed an enhanced survival rate, compared with the control group of mice (Figure 5D and Appendix A). These results suggest that DRD3-signalling in CD8^+^ T-cells provides a higher anti-tumour potency in vivo.

## 4. Discussion

This study provides genetic evidence indicating that DRD3 signalling reinforces the production of IL-2 and the expression of the high-affinity IL-2 receptor in CD8^+^ T-cells, which are associated with higher expansion and acquisition of effector function by these lymphocytes, and with enhanced efficacy of the anti-tumour response in a melanoma mouse model. Thus, these findings contribute to basic knowledge of how dopaminergic signalling regulates the T-cell response against melanoma and also represent an interesting opportunity for melanoma therapy.

Here, we found that DRD3 signalling on CD8^+^ T-cells potentiates the anti-tumour immune response in a melanoma mouse model. These findings agree with the hypothesis that stimulating high-affinity dopamine receptors enhances the inflammatory response whilst activating low-affinity dopamine receptors induces anti-inflammatory mechanisms. In support of this, a study performed with human samples obtained from head and neck cancer showed that 10 nM dopamine (which should stimulate DRD3 selectively) increases the migration of T-cells towards autologous tumours [36]. Moreover, studies performed with other high-affinity dopamine receptors, including DRD4 and DRD5 ^4^, have also shown a pro-inflammatory effect mediated through these receptors. A recent study demonstrated that signalling through DRD5 promotes the differentiation of CD8^+^ T-cells into tissue-resident memory T-cells, which are associated with a robust protective anti-tumour immune response [37]. Another study recently demonstrated that signalling through DRD4 favours the Th2-mediated allergic inflammation of the lungs in young individuals [6]. Furthermore, a group of studies has consistently demonstrated that stimulating the low-affinity receptor DRD1 in myeloid cells attenuates the activation of the inflammasome NLRP3, thereby dampening inflammation in several contexts, including neuroinflammation, lipopolysaccharide-induced systemic inflammation, and acute kidney injury [38,39,40,41]. Also consistent with this general hypothesis, it has been shown that stimulation of the low-affinity receptor DRD2 in astrocytes attenuates neuroinflammation in a mouse model of Parkinson’s disease [42]. Nevertheless, a few results are contradictory to this hypothesis. For instance, our previous study indicated that DRD3 signalling in DCs reduces the presentation of tumour-derived antigens to CD8^+^ T-cells, thereby reducing the priming of tumour-specific lymphocytes and attenuating the anti-tumour immune response [24]. In addition, a previous work addressing the effect of DRD3 stimulation in the function of NK cells provided pharmacologic evidence indicating that DRD3 signalling inhibits the cytotoxic activity of splenic NK cells in vitro [43]. Additionally, recent research has shown pharmacologic and genetic evidence indicating that signalling through the high-affinity receptor DRD5 attenuates neutrophil migration towards the infection site [44]. Therefore, the general hypothesis that low/high-affinity dopamine receptors mediate anti/pro-inflammatory mechanisms does not always work. Further research in this area is necessary to better understand how dopaminergic signalling shapes the function of the immune system.

Previous evidence has shown that dopamine may affect cancer development through other dopamine receptors, different from DRD3, by acting directly on tumour cells or indirectly on cells associated with the tumour microenvironment [45]. The direct effect of dopamine on tumour cells is supported by studies in which dopaminergic drugs targeting DRD1 and DRD2 exert therapeutic effects in preclinical cancer models by inhibiting tumour proliferation and inducing death of tumour cells [45,46].

Regarding the indirect effects of dopamine on tumour development, a group of studies has indicated that DRD2 signalling is a potent inhibitor of angiogenesis in several pre-clinical cancer models [47,48,49,50,51]. Of note, angiogenesis, which corresponds to the formation of new blood vessels, constitutes an essential process to support tumour growth and progression [52,53]. Mechanistic analyses have showed that the therapeutic effect of DRD2 signalling in cancer was mediated through the inhibition of the signalling triggered by the vascular endothelial growth factor (VEGF) on endothelial cells [48,51]. Of note, the DRD2-mediated inhibition of angiogenesis also reduces the extent of the infiltration of immune suppressive cells, such as myeloid-derived suppressor cells (MDSC) [47].

Indirect mechanisms of dopaminergic regulation of tumour growth also include the effects of dopamine on immune cells. In this regard, macrophages, DCs, and MDSC represent essential components of the tumour microenvironment that determine tumour growth and progression. Liu and colleagues have found that, through DRD4 stimulation, dopamine dampens the acquisition of the M2 phenotype by tumour-associated macrophages, a subset of macrophages that support tumour development [54]. Accordingly, the same authors have shown that DRD4 signalling on macrophages improves the efficacy of gemcitabine chemotherapy in mouse models of pancreatic cancer [54]. Similarly, the DRD2 agonist aripiprazole has been shown to increase the sensitivity of pancreatic cancer to chemotherapy [55]. Analogously, a study carried out in an orthotopic rat model of glioma provided evidence indicating that DRD2 signalling induces a shift in the functional phenotype of macrophages from M2 to M1 and attenuates tumour growth [51]. The dopaminergic regulation of MDSC has also been shown to exert relevant effects on tumour growth. For instance, Wu and colleagues have shown that the pharmacologic stimulation of type I dopamine receptors (including DRD1 and DRD5) reduces the suppressive function of MDSC, thus enhancing the potency of the T-cell-mediated anti-tumour response [56]. Dopaminergic pathways of the CNS also regulate the suppressive activity of MDSC involved in tumour development. Accordingly, previous studies addressing the role of dopaminergic neural pathways in the control of immunity and cancer have shown that the activity of the reward system increases the innate and adaptive immune response that limits tumour growth [57,58]. Mechanistic analyses have showed that the dopaminergic neurons of the ventral tegmental area (VTA) in the CNS triggered the activity of sympathetic inputs in the bone marrow, dampening the immunosuppressive function of MDSC through the stimulation of noradrenergic receptors [58].

In a previous study, Watanabe and colleagues conducted experiments in which the i.p. injection of dopamine induced chemotaxis of naïve CD8^+^ T-cells towards the intraperitoneal cavity, an effect abrogated by the selective DRD3 antagonist U-99194A [25]. In the same study, the authors showed that, 24 h after the i.p. injection of 20 mg/Kg of U-99194A, the infiltration of naïve CD8^+^ T-cells in the inguinal lymph nodes was decreased [25]. Moreover, when CCR7 was desensitised on CD8^+^ T-cells by ex vivo incubation with CCL19, the effect of the i.p. injection of U-99194A and CCR7 desensitisation were synergistic in the reduction of naïve CD8^+^ T-cells into the inguinal lymph nodes [25]. Since presenting tumour-derived antigens by APCs to tumour-specific CD8^+^ T-cells in the lymph nodes is a fundamental step to trigger a proper anti-tumour immune response, we evaluated whether *Drd3*-deficiency of tumour-specific CD8^+^ T-cells affected their migration through the secondary lymphoid organs. Our results provided genetic evidence that DRD3 neither affected the infiltration of these cells into the secondary lymphoid organs nor changed the surface expression of CCR7 and CD62L (Figure 2), which correspond to the homing molecules leading the migration of T-cells through secondary lymphoid organs [59]. The discrepancy between our results and those of Watanabe and colleagues could be due to the fact that they injected 20 mg/kg U-99194A i.p. in their experimental mice, a concentration that may stimulate not only DRD3 (ki = 160 nM for U-99194A), but also DRD2 (ki = 2281 nM for U-99194A) in vivo [60].

Interestingly, similar to the effect of DRD3 on CD8^+^ T-cells observed here, our previous studies showed that DRD3 stimulation on CD4^+^ T-cells strengthens the production of IL-2 and also potentiates the differentiation towards the Th1 effector phenotype and the consequent production of IFN-γ [11,17]. Thus, it is likely that DRD3 signalling on CD4^+^ T-cells also potentiates the anti-tumour immune response. Moreover, DRD3 stimulation has been proposed to increase the M1 inflammatory profile in macrophages [15], a subset that contributes to the anti-tumour response. The suppressive activity of Treg, which is detrimental to anti-tumour immunity [61], is also inhibited by DRD3 signalling [10]. Therefore, considering the evidence describing the functional effects of DRD3 signalling on effector CD4^+^ T-cells, Treg, macrophages, and the results obtained here in CD8^+^ T-cells, it is tempting to speculate that treatment with selective DRD3 agonists should exert a therapeutic effect in tumour development. Nevertheless, although the effect of DRD3 stimulation in T-cells and macrophages is expected to reinforce the anti-tumour immune response, DRD3 signalling on DCs [24] and NK cells [43] is probably harmful to anti-tumour immunity. Therefore, the therapeutic potential of targeting DRD3 in cancer should be experimentally evaluated in future studies.

## Figures and Tables

**Figure 1 cells-11-03536-f001:**
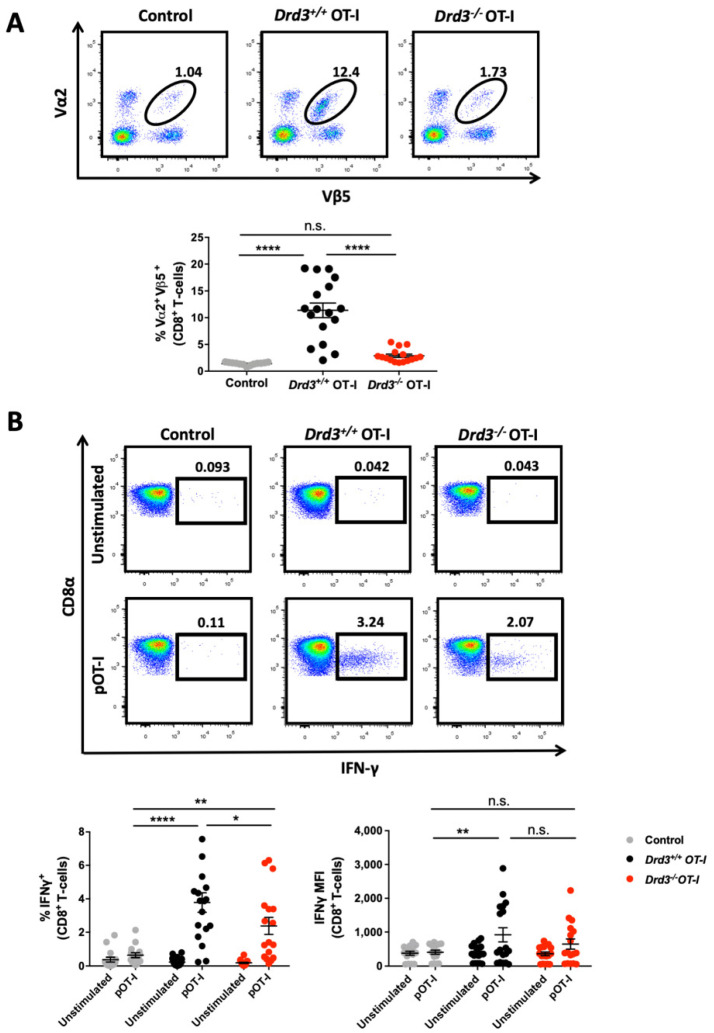
*Drd3*-deficiency results in attenuated CD8^+^ T-cell expansion and reduced IFN-γ production. Splenic CD8^+^ T-cells were isolated from *Drd3^+/+^* or *Drd3^−/−^* OT-I mice, and then i.v. transferred into male or female C57BL/6 mice (2 × 10^5^ cells/mouse). A group of mice did not receive the transfer of exogenous T-cells (Control). One day later, mice received an intradermal vaccination with 40μg of pVAX-OVA and, 11 days after vaccination, blood samples were obtained. (**A**) The expansion of OT-I cells was evaluated as the percentage of cells expressing the transgenic TCR (Vα2 Vβ5) in the CD3^+^ CD8^+^ ZAq^−^ gate. Representative dot-plots are shown in the top panel. Numbers indicate the percentage of cells in the indicated region. Quantification is shown in the bottom panel. (**B**) Cells were left unstimulated or stimulated with pOT-I (1 μg/mL) for 6 h, and the intracellular production of IFNγ was evaluated in the CD8^+^ T-cell population by flow cytometry. Representative dot-plots are shown in the top panel. Numbers indicate the percentage of cells in the indicated region. Quantification of the percentage of IFNγ^+^ CD8^+^ T-cells (left bottom panel) and the mean fluorescence intensity (MFI) associated with IFNγ^+^ in the CD8^+^ ZAq^−^ gate are shown. (**A**,**B**) Data from three independent experiments (*n* = 15–18 mice per group) is shown. Each symbol represents data from an individual mouse. The mean ± SEM are depicted. *, *p* < 0.05; **, *p* < 0.01; ****, *p* < 0.0001 by one-way (**A**) or two-way (**B**) ANOVA followed by Tukey’s post hoc test. n.s., not significant.

**Figure 2 cells-11-03536-f002:**
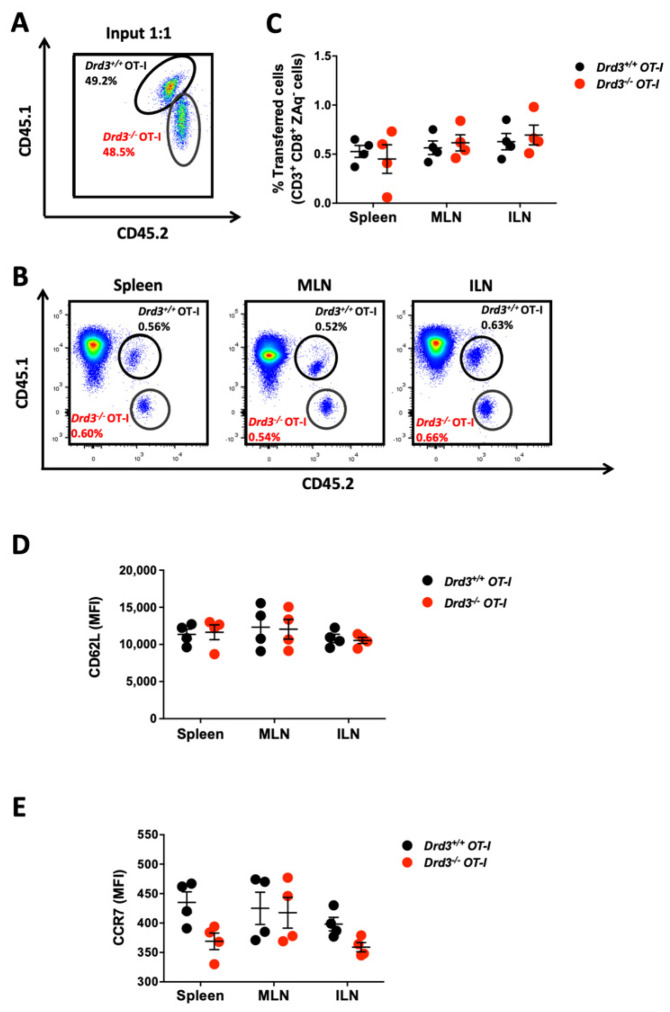
*Drd3*-deficiency does not affect CD8^+^ T-cell migration through secondary lymphoid organs. Splenic CD8^+^ T-cells were isolated from *Drd3^+/+^ Cd45.1^+/−^ CD45.2^+/−^* or *Drd3^−/−^ Cd45.1^−/−^ CD45.2^+/+^* OT-I mice and then mixed in a 1:1 ratio and i.v. transferred into *Cd45.1^+/+^ CD45.2^−/−^* male C57BL/6 mice (2 × 10^6^ total cells/mouse). Twenty-four hours later, inguinal lymph nodes (ILN), mesenteric lymph nodes (MLN), and the spleen were collected, and CD8^+^ T-cells were analysed by flow cytometry. (**A**) A representative dot-plot of the input of 1:1 mixed donors’ cells is shown. (**B**) Representative dot-plots analysing the proportion of *Drd3^+/+^* (CD45.1^+^ CD45.2^+^) and *Drd3^−/−^* (CD45.1^−^ CD45.2^+^) donor’s OT-I cells in the CD8^+^ ZAq^−^ gate from different secondary lymphoid organs are shown. (**A,B**) Numbers indicate the percentage of cells in the indicated region. (**C**) Quantification of the percentage of *Drd3^+/+^* and *Drd3^−/−^* donors’ OT-I cells in the CD8^+^ ZAq^−^ gate. (**D**,**E**) Quantification of the MFI associated with CD62L (**D**) and CCR7 (**E**) expression on *Drd3^+/+^* and *Drd3^−/−^* donors’ OT-I cells infiltrating different secondary lymphoid organs are shown. (**C**–**E**) Each symbol represents data from an individual mouse. Data from two independent experiments is shown. The mean ± SEM are depicted. No significant differences were found by two-way ANOVA.

**Figure 3 cells-11-03536-f003:**
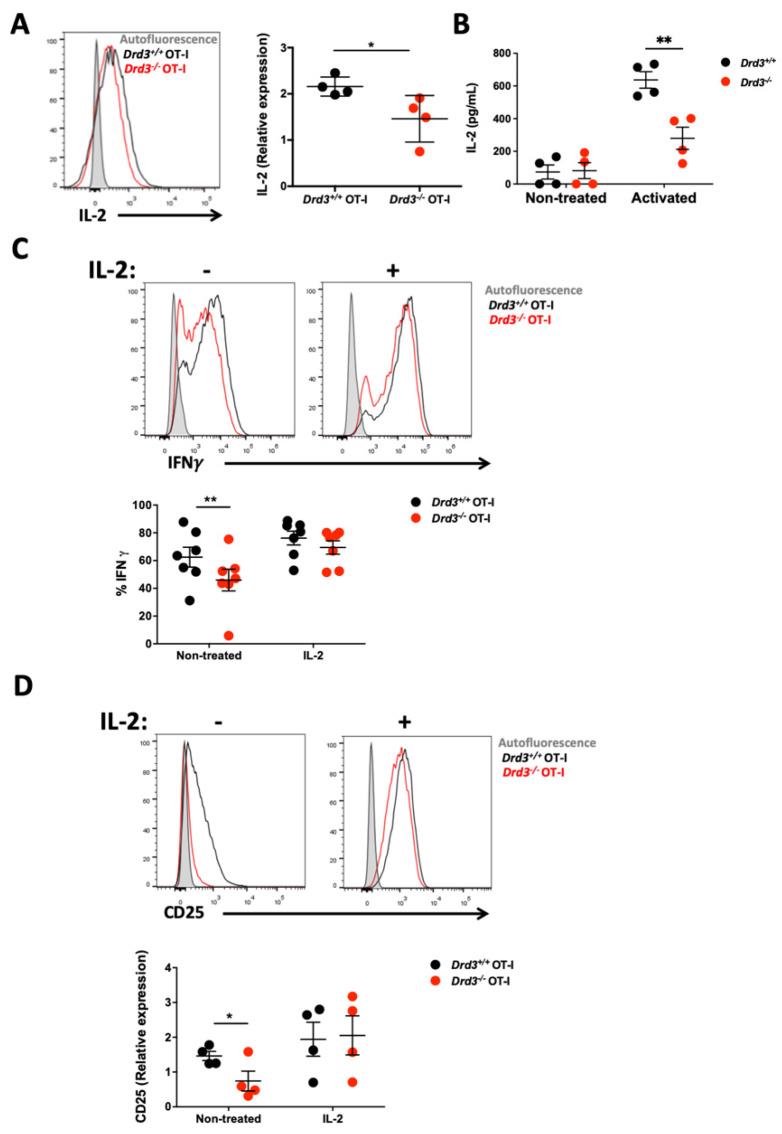
DRD3-signalling strengthens IFN-γ production by favouring IL-2 release. (**A**) Splenic CD8^+^ T-cells were isolated from *Drd3^+/+^* or *Drd3^−/−^* OT-I male mice and stimulated with anti-CD3 and anti-CD28 Abs in the absence of exogenous IL-2 for 48 h. Subsequently, cells were cultured for an additional 48 h in the absence of anti-CD3 and anti-CD28 Abs. The pOT-I was added to the cultures for the last 6 h, and then IL-2 was evaluated by intracellular immunostaining followed by flow cytometry analysis. The left panel shows representative histograms. The right panel shows the quantification. Values are the MFI associated with IL-2 in the CD8^+^ ZAq^−^ gate normalised to the value of IL-2 MFI obtained for non-stimulated OT-I cells. (**B**) Splenic CD8^+^ T-cells isolated from *Drd3^+/+^* or *Drd3^−/−^* male mice were non-treated or activated with anti-CD3 and anti-CD28 Abs in the absence of exogenous IL-2 for 24 h, and the extent of IL-2 secretion into the supernatant was quantified by ELISA. Each symbol represents data obtained from an individual mouse. (**C**,**D**) *Drd3^+/+^* or *Drd3^−/−^* OT-I CD8^+^ T-cells were activated with anti-CD3 and anti-CD28 Abs in the absence or the presence of exogenous IL-2 during 48 h. Cells were cultured for an additional 48 h in the absence of anti-CD3 and anti-CD28 Abs. The pOT-I was added to the cultures for the last 6 h, and the intracellular IFNγ production (**C**) and surface CD25 expression (**D**) were assessed by flow cytometry. (**C**,**D**) The top panels show representative histograms and the bottom panels the quantification of the percentage of IFNγ^+^ cells (**C**) and the normalised MFI for CD25 (**D**) in the CD8^+^ ZAq^−^ gate. The histograms associated with autofluorescence are shown as a reference in the top panels. (**A**–**D**) Each symbol represents data obtained from a different mouse. Data from four (**A**,**B**,**D**) or seven (**C**) independent experiments is shown. The mean ± SEM are depicted. *, *p* < 0.05; **, *p* < 0.01; by Student’s *t*-test (**A**) or two-way ANOVA followed by Tukey’s post hoc test (**B**,**C**).

**Figure 4 cells-11-03536-f004:**
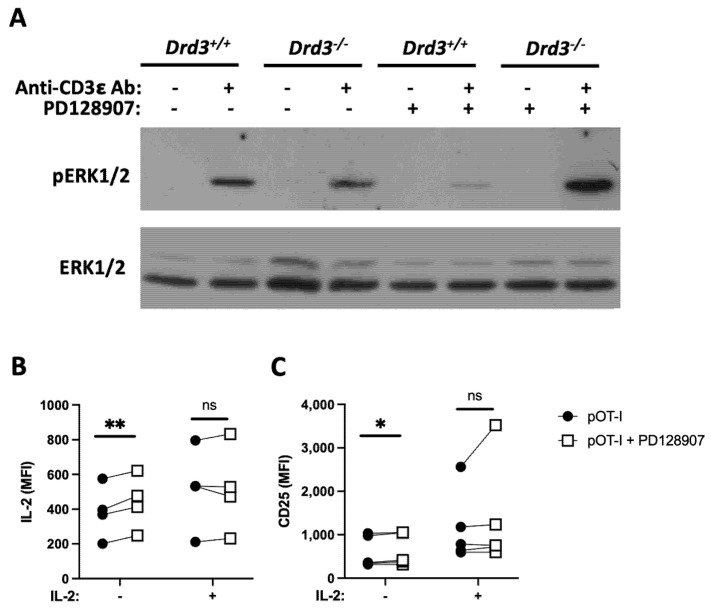
DRD3 stimulation reduces ERK1/2 phosphorylation and increases IL-2 production triggered by the TCR activation. (**A**) Splenic *Drd3^+/+^* or *Drd3^−/−^* CD8^+^ T-cells were purified and then left unstimulated or stimulated with either anti-CD3 Ab, PD128907, or both together for 5 min. Afterwards, cells were lysed and the extent of ERK1/2-phosphorylation (pERK1/2; top panel) or total ERK1/2 (ERK1/2; bottom panel) were analysed by western blot. Results from a representative of three independent experiments are shown. (B-C) *Drd3^+/+^* OT-I CD8^+^ T-cells were left untreated or treated with 50 nM PD128907 for 30 min and then activated in the presence or absence of exogenous IL-2 (as described in Section 2.3.). The intracellular IL-2 production (**B**) and surface CD25 expression (**C**) were assessed by flow cytometry. Values are the MFI obtained in four independent experiments. Each symbol represents data obtained from a different mouse. Data from four independent experiments are shown. *, *p* < 0.05; **, *p* < 0.01; by paired Student’s *t*-test.

**Figure 5 cells-11-03536-f005:**
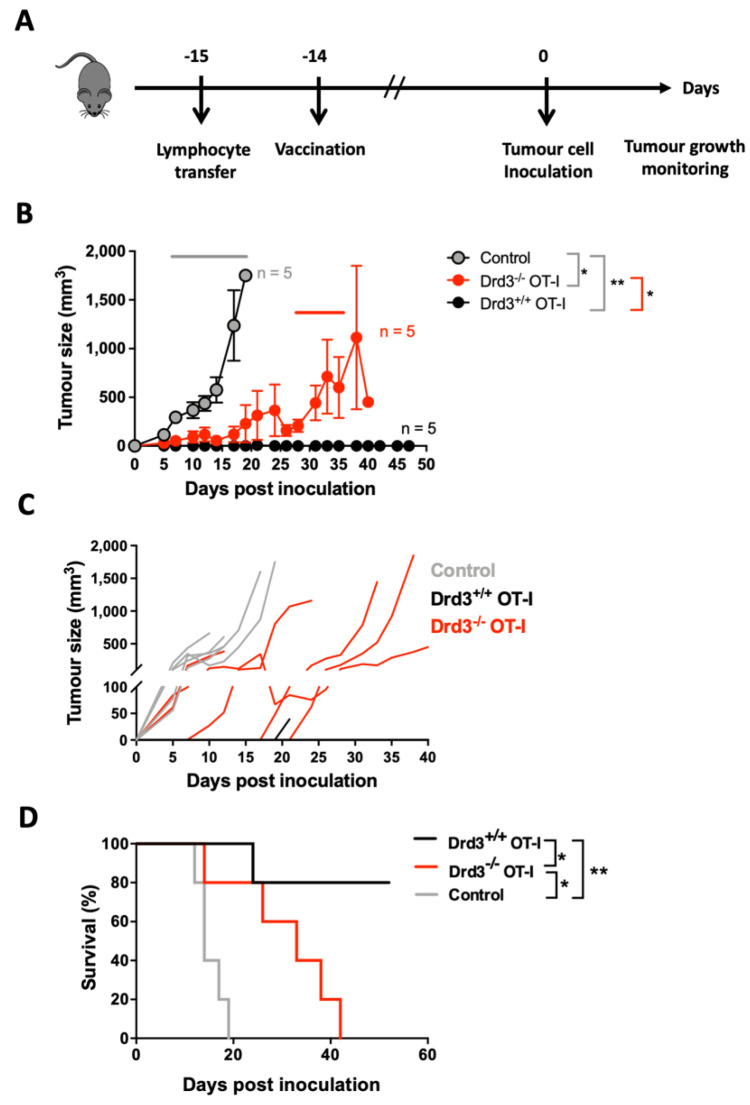
*Drd3*-deficient CD8^+^ T-cells exhibit reduced potency in the anti-tumour response in male mice. Splenic CD8^+^ T-cells were isolated from *Drd3^+/+^* or *Drd3^−/−^* OT-I mice, and then i.v. transferred into male C57BL/6 mice (2 × 10^5^ cells/mouse). A group of mice did not receive the transfer of exogenous T-cells (Control). One day later, mice received an intradermal vaccination with 40 μg of pVAX-OVA. Thirteen days after vaccination, mice were s.c. inoculated with B16/OT-I melanoma cells (10^6^ cells/mouse), and tumour growth and mice survival were monitored over time. (**A**) The experimental design is illustrated. (**B**) Tumour growth is represented as tumour volume in time. Data are the mean ± SEM from five mice per group. The top grey line indicates the frame of time where significant differences were found compared to the control group, and the red line indicates the frame of time where significant differences were found between mice receiving *Drd3^+/+^* or *Drd3^−/−^* OT-I cells. (**C**) Tumour growth curves for individual mice are shown. (**D**) Mice’s survival over time is shown. Data from a representative (n = 5 mice per group) of two independent experiments are shown. *, *p* < 0.05; **, *p* < 0.01 by two-way ANOVA followed by Tukey’s post hoc test (B, grey line), by multiple t-test (B, red line) or by logrank test (**D**). n.s., not significant.

## Data Availability

The datasets used and/or analysed during the current study are available from the corresponding author on reasonable request.

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
