# Peer review of "Dopaminergic Signalling Enhances IL-2 Production and Strengthens Anti-Tumour Response Exerted by Cytotoxic T Lymphocytes in a Melanoma Mouse Model"

_cells, 2022, doi:10.3390/cells11223536_

Round 1

Reviewer 1 Report

This manuscript, Chovar et. al., addresses a topic of increasing interest and importance – the immunologic and pathogenic impact of dopamine. This study specifically addresses the role of dopamine receptor 3 (DRD3) in the CTL mediated anti-tumor response. The Pacheco lab is well-versed in this type of research, having published a number of notable studies examining the impact of dopamine, particularly activation of DRD3 and DRD5 receptors. This study uses a similar approach, identifying the role of DRD3 in anti-tumor activity in CD8+ T-cells via transfer of immune cells with genetically knocked out DRD3. This is an excellent approach to this type of study, as it enables identification of the specific receptors involved in the process being studied. Chovar and colleagues use this technique to good effect here, as their data indicate that the presence of DRD3 on transferred CD8+ T-cells is important to the in vivo expansion of CTLs. This receptor is also important to the production of IFN-y by CTL in response to IL-2 via CD25. Overall, this study adds substantially to the field of dopaminergic immunology, demonstrating a specific role of DRD3 in the anti-tumor response mediated by CD8+ T cells.  However, there are a number of changes that could be made to strengthen these conclusions and improve the accessibility of this study. These changes are discussed below. 

1.    The authors describe differences between the high affinity/low affinity dopamine receptor involvement in inflammatory vs anti-inflammatory responses, but only present one perspective on this topic. While a number of studies from this group have shown that high-affinity dopamine receptors may be proinflammatory and low-affinity anti-inflammatory, the data on this topic are mixed, and a number of studies do not support this conclusion. As dopaminergic immunology is a relatively new field and is not particularly well understood, it would be helpful to include some additional discussion and references to explain this concept, as well as the caveats and opposing viewpoints from other studies.

2.    Knocking out DRD3 is a very clean way to target this receptor, but it could also lead to other effects via expanded activation of non-DRD3 receptors on T-cells now that the highest affinity receptor is gone. Some consideration of the knock-on effects of DRD3 knock-out on other receptors, and just generally consideration of whether other receptors are at all involved in this process, and why or why not, would be beneficial. 

3.    For people who are not cancer biologists, the experimental design in this manuscript is relatively opaque. Additional description of the model, i.e., what is the expected outcome of transferring peptide specific CD8+ T-cells, why does using an OVA vaccine mediate this outcome, why are OVA-specific OT-I transgenic mice used and why do you expect to see changes in the Va2+ Vb5+ CD8+ T-cells? Additionally, a cartoon or figure showing the experimental process and the timeline of the results found would be extremely helpful. 

4.    Although not all in melanoma, there have been a number of studies examining the role of dopamine in anti-tumor responses that could be included to better situate the data regarding what is known in this area. For example, PMIDs 28928888, 21047943, 25030361, 25818600 and others, speak directly to this topic, and could be included to improve the placement of these data in the context of dopamine and cancer. 

5.    In Figure 1, the expansion of Va2+ Vb5+ CD8+ T-cells in mice with Drd3-deficient OT-I was similar to the expansion of endogenous Va2+ Vb5+ CD8+ T-cells in control mice that were not immunized. However, the percentage of IFN-y producing cells was different between these two groups. This discrepancy should be explained.

6.    There are some additional experiments that substantially improve this study, and  the use of additional modalities or experimental methods to valid the existing findings would also greatly strengthen the explanations used in this proposal. 

a.    For measurement of cytokines, use of ELISAs (or equivalent) to measure IL-2 and IFN-y would be very helpful. This would also address a minor problem with the data interpretation in this manuscript, i.e., the comments that the experiments measure cytokine secretion. Flow cytometry either measures the number of cells containing the antigen being stained for or the average intensity of antigen staining in those cells. Neither of these processes actually address secretion of cytokines, so this language should be changed, and the discussion of the results modified to specifically note what is being examined. 

b.    For measurement of surface receptors (CD25, CD62L) Western blots  of the specific T-cell population being examined would be a useful confirmatory experiment. 

c.     Using pharmacological ligands (agonist and agonist + antagonist) that are specific to DRD3 in WT animals injected with OVA-specific OT-I would be a strong validation of the role of DRD3. 

d.    Additional interrogation of the signaling pathways or transcription factors connecting DRD3 with CD25, IL-2 signaling with IFN-y production, and all of these mechanisms with anti-tumor activity would be very helpful. The experiments in this manuscript are fairly complex but well-done, however, they are not specifically linked and the posited connections between them may be a bit overinterpreted and need additional study to solidify the proposed interaction. For example, showing the activity of signaling intermediates that connect these processes, or testing the effects of disrupting / enhancing these processes on tumor growth and survival. 

7.    Please clarify if the mice used were either male or female, or if both sexes were used, and whether there were any sex differences in the results. Even if these differences were not significant, it would be worth noting trends that differ between sexes as this could suggest future studies for the broader field. 

8.    The discussion of astrocytes in lines 61 – 63 is surprising and a bit out of context. There are a substantial number of studies associated with the inflammatory impact of DRD3, and its effects in astrocytes seems less germane to this topic than a number of other potential references. 

9.    The figures could be at a higher resolution as it is a little difficult to read some of the axis titles and legends. In Figure 4B and C it is hard to tell if there was any tumor growth at all in the DRD3 sufficient mice. If there wasn’t this should be indicated. If there was, could the y axis be split to show the tumor growth (albeit at a much smaller magnitude)? 

10. There are a number of grammatical, syntactical and repetition errors that could be corrected with more through editing. For example, “In adittion, CTL may promote the anti-tumour activity of macrophages …” on line 84, in the Figure 4 legend “Tumour growth is represented as tumour volume in the time”, in line 272 “how was the extent of IL-2 production”, and in lines 285-287 “A critical step on the action mechanism of IL-2 during T-cell expansion is the autocrine upregulation of the aIL-2 receptor (CD25), which increases the affinity of the IL-2 receptor about 100 folds, making T-cells substantially 287 more sensitive to IL-2”. Language could also be more precise. For example, in lines 65-66 “but also in some organs where dopamine just sometimes appears and in low levels.” 

Author Response

Reviewer 1:

Comments and Suggestions for Authors

This manuscript, Chovar et. al., addresses a topic of increasing interest and importance – the immunologic and pathogenic impact of dopamine. This study specifically addresses the role of dopamine receptor 3 (DRD3) in the CTL mediated anti-tumor response. The Pacheco lab is well-versed in this type of research, having published a number of notable studies examining the impact of dopamine, particularly activation of DRD3 and DRD5 receptors. This study uses a similar approach, identifying the role of DRD3 in anti-tumor activity in CD8+ T-cells via transfer of immune cells with genetically knocked out DRD3. This is an excellent approach to this type of study, as it enables identification of the specific receptors involved in the process being studied. Chovar and colleagues use this technique to good effect here, as their data indicate that the presence of DRD3 on transferred CD8+ T-cells is important to the in vivo expansion of CTLs. This receptor is also important to the production of IFN-y by CTL in response to IL-2 via CD25. Overall, this study adds substantially to the field of dopaminergic immunology, demonstrating a specific role of DRD3 in the anti-tumor response mediated by CD8+ T cells.  However, there are a number of changes that could be made to strengthen these conclusions and improve the accessibility of this study. These changes are discussed below. 

  1. The authors describe differences between the high affinity/low affinity dopamine receptor involvement in inflammatory vs anti-inflammatory responses, but only present one perspective on this topic. While a number of studies from this group have shown that high-affinity dopamine receptors may be proinflammatory and low-affinity anti-inflammatory, the data on this topic are mixed, and a number of studies do not support this conclusion. As dopaminergic immunology is a relatively new field and is not particularly well understood, it would be helpful to include some additional discussion and references to explain this concept, as well as the caveats and opposing viewpoints from other studies.

ANSWER: We thank the reviewer for this comment. We have included a discussion about this issue in the second paragraph of the discussion in the new version of the paper.

  1. Knocking out DRD3 is a very clean way to target this receptor, but it could also lead to other effects via expanded activation of non-DRD3 receptors on T-cells now that the highest affinity receptor is gone. Some consideration of the knock-on effects of DRD3 knock-out on other receptors, and just generally consideration of whether other receptors are at all involved in this process, and why or why not, would be beneficial. 

ANSWER: To confirm the conclusion about the role of DRD3 in CD8+ T-cells obtained with knockout cells, we performed new experiments using a pharmacologic approach. In these experiments, we observed that the stimulation of wild-type CD8+ T-cells with a DRD3-selective agonist induced a potentiation in the production of IL-2 and the surface expression of CD25. These effects were abrogated in the presence of exogenous IL-2 added to the culture medium. Thus, these new experiments confirm our conclusions about the role of DRD3-mediated effects in CD8+ T-cells, but now using wild-type cells, thereby avoiding the potential effects of Drd3deficiency on the expression/effects of other dopamine receptors. These new results are shown in the figure 4 of the new version of the paper. 

  1. For people who are not cancer biologists, the experimental design in this manuscript is relatively opaque. Additional description of the model, i.e., what is the expected outcome of transferring peptide specific CD8+ T-cells, why does using an OVA vaccine mediate this outcome, why are OVA-specific OT-I transgenic mice used and why do you expect to see changes in the Va2+ Vb5+ CD8+ T-cells? Additionally, a cartoon or figure showing the experimental process and the timeline of the results found would be extremely helpful. 

ANSWER: A more detailed explanation of the model has been written in the section 2.1. Moreover, we have included a new figure (figure S1) where an illustration of the model and a timeline of the experimental design were included to improve the clarity of our study.

  1. Although not all in melanoma, there have been a number of studies examining the role of dopamine in anti-tumor responses that could be included to better situate the data regarding what is known in this area. For example, PMIDs 28928888, 21047943, 25030361, 25818600 and others, speak directly to this topic, and could be included to improve the placement of these data in the context of dopamine and cancer. 

ANSWER: The indicated references were included in the second, third, fourth, and fifth paragraphs of the discussion in the new version of the paper.

  1. In Figure 1, the expansion of Va2+ Vb5+ CD8+ T-cells in mice with Drd3-deficient OT-I was similar to the expansion of endogenous Va2+ Vb5+ CD8+ T-cells in control mice that were not immunized. However, the percentage of IFN-y producing cells was different between these two groups. This discrepancy should be explained.

      ANSWER: Although the expansion of Drd3-deficient OT-I CD8+ T-cells was similar to the extent of expansion of endogenous Va2+ Vb5+CD8+ T-cells present in control mice (Figure 1A), the group of Drd3-deficient OT-I CD8+ T-cells contained a higher percentage of cells producing IFN-g in response to pOT-I in comparison to the control group (Figure 1B). This discrepancy might be due to that the extent of expansion was analysed in the Va2+ Vb5+ CD8+ gate (Figure 1A), whilst the IFN-g production was determined in the total CD8+population in response to the ex vivo restimulation with pOT-I (Figure 1B). Thereby, it is possible that some cells producing IFN-gdetected in the Drd3-deficient group correspond to endogenous pOT-I specific CD8+ T-cells. Although this fact may induce an overvalue in the percentage of cells producing IFN-g in the Drd3-deficient group, this does not change the main conclusion of these experiments, which indicates that the percentage of cells producing IFN-g is lower in Drd3-deficient OT-I CD8+ T-cells compared with Drd3-sufficient OT-I CD8+ T-cells. This discussion has been included at the end of section 3.1. in the new version of the paper.

  1. There are some additional experiments that substantially improve this study, and  the use of additional modalities or experimental methods to valid the existing findings would also greatly strengthen the explanations used in this proposal. 

  1. For measurement of cytokines, use of ELISAs (or equivalent) to measure IL-2 and IFN-y would be very helpful. This would also address a minor problem with the data interpretation in this manuscript, i.e., the comments that the experiments measure cytokine secretion. Flow cytometry either measures the number of cells containing the antigen being stained for or the average intensity of antigen staining in those cells. Neither of these processes actually address secretion of cytokines, so this language should be changed, and the discussion of the results modified to specifically note what is being examined. 

ANSWER: According to the reviewer’s recommendations, we conducted new experiments in which we confirmed the primary effect of Drd3-deficiency in CD8+ T-cells, the impaired IL-2 production, but at the level of cytokine secretion in the supernatant. These new results obtained by ELISA show that the Drd3-deficiency substantially impairs the secretion of IL-2 by CD8+ T-cells.. These new results are included in the figure 3B in the new version of the paper.

  1. For measurement of surface receptors (CD25, CD62L) Western blots  of the specific T-cell population being examined would be a useful confirmatory experiment. 

ANSWER: The antibodies used to detect CD25 and CD62L corresponds to monoclonal antibodies that have been validated in more than 10 and 60 studies respectively. Moreover, each lot of these monoclonal antibodies are quality control tested by immunofluorescent immunostaining using negative and positive controls. For these reasons, we think western blots are not necessary to confirm the specificity of these monoclonal antibodies.

For more information of the anti-CD25 antibody and the references associated please visit the link:

https://d1spbj2x7qk4bg.cloudfront.net/fr-ch/products/fitc-anti-mouse-cd25-antibody-4511?pdf=true&displayInline=true&leftRightMargin=15&topBottomMargin=15&filename=FITC%20anti-mouse%20CD25%20Antibody.pdf&v=20171116053513

For more information of the anti-CD62L antibody and the references associated please visit the link:

https://d1spbj2x7qk4bg.cloudfront.net/fr-ch/products/apc-anti-mouse-cd62l-antibody-381?pdf=true&displayInline=true&leftRightMargin=15&topBottomMargin=15&filename=APC%20anti-mouse%20CD62L%20Antibody.pdf&v=20220831123135

  1. Using pharmacological ligands (agonist and agonist + antagonist) that are specific to DRD3 in WT animals injected with OVA-specific OT-I would be a strong validation of the role of DRD3. 

ANSWER: To gain more robust evidence in the mechanistic analysis of our study, we confirmed our results obtained with Drd3-deficient CD8+ T-cells, but using a pharmacologic approach in vitro. To this end, we used a DRD3-selective agonist, PD128907. We confirmed that, in the absence of exogenous IL-2, the treatment of Drd3+/+ OT-I CD8+ T-cells with the DRD3-agonist exacerbated the production of IL-2 and the surface expression of CD25. Of note, the differences in IL-2 production and CD25 expression observed between PD128907-treated and untreated Drd3+/+ OT-I CD8+ T-cells were abolished in the presence of exogenous IL-2. These new results are included in the figure 4B-C in the new version of the paper.

  1. Additional interrogation of the signaling pathways or transcription factors connecting DRD3 with CD25, IL-2 signaling with IFN-y production, and all of these mechanisms with anti-tumor activity would be very helpful. The experiments in this manuscript are fairly complex but well-done, however, they are not specifically linked and the posited connections between them may be a bit overinterpreted and need additional study to solidify the proposed interaction. For example, showing the activity of signaling intermediates that connect these processes, or testing the effects of disrupting / enhancing these processes on tumor growth and survival. 

      ANSWER: To address this issue, we performed new experiments analyzing how DRD3 stimulation modifies intracellular signalling pathways in CD8+ T-cells. The ERK1/2 is one signalling pathway triggered by the TCR-activation modulated by the DRD3-stimulation in CD4+ T-cells. Thereby, we evaluated how a selective DRD3-agonist (PD128907) affected the phosphorylation of ERK1/2 in TCR-stimulated Drd3+/+ and Drd3-/- CD8+ T-cells. The results show that DRD3-stimulation reduced the ERK1/2-phosphorylation induced by the TCR-activation in Drd3-sufficient CD8+ T-cells, an effect abrogated in Drd3-deficient CD8+ T-cells. Of note, in the absence of TCR-activation, the DRD3 stimulation did not affect ERK1/2-phosphorylation in either Drd3+/+ or Drd3-/- CD8+ T-cells. These new results are included in figure 4A in the new version of the paper.

  1. Please clarify if the mice used were either male or female, or if both sexes were used, and whether there were any sex differences in the results. Even if these differences were not significant, it would be worth noting trends that differ between sexes as this could suggest future studies for the broader field. 

ANSWER: In the new version of the paper, we included supplementary figures analysing the in vivo expansion of  OT-I CD8+ T-cells (Figure S2) and the in vivo production of IFN-g by OT-I CD8+ T-cells (Figure S3) separated by gender. In addition, we also provided the results of the anti-tumour response mediated by Drd3-sufficient or Drd3-deficient OT-I CD8+ T-cells separately for males (Figure 5) and females (figure S4) in the new version of the paper.

  1. The discussion of astrocytes in lines 61 – 63 is surprising and a bit out of context. There are a substantial number of studies associated with the inflammatory impact of DRD3, and its effects in astrocytes seems less germane to this topic than a number of other potential references. 

ANSWER: According to the reviewer’s recommendation, the comments about the pro-inflammatory role of DRD3-signalling on astrocytes has been removed from the introduction. 

  1. The figures could be at a higher resolution as it is a little difficult to read some of the axis titles and legends. In Figure 4B and C it is hard to tell if there was any tumor growth at all in the DRD3 sufficient mice. If there wasn’t this should be indicated. If there was, could the y axis be split to show the tumor growth (albeit at a much smaller magnitude)? 

ANSWER: The resolution of the figures has been improved in the new version of the paper. Regarding the comment about Figures 4B and 4C (Figures 5B and 5C in the new version of the paper), we included a split Y-axis in the graph shown in C to appreciate better a small degree of tumour growth in one of the mice belonging to the group of mice receiving Drd3+/+ OT-I cells.

  1. There are a number of grammatical, syntactical and repetition errors that could be corrected with more through editing. For example, “In adittion, CTL may promote the anti-tumour activity of macrophages …” on line 84, in the Figure 4 legend “Tumour growth is represented as tumour volume in the time”, in line 272 “how was the extent of IL-2 production”, and in lines 285-287 “A critical step on the action mechanism of IL-2 during T-cell expansion is the autocrine upregulation of the aIL-2 receptor (CD25), which increases the affinity of the IL-2 receptor about 100 folds, making T-cells substantially 287 more sensitive to IL-2”. Language could also be more precise. For example, in lines 65-66 “but also in some organs where dopamine just sometimes appears and in low levels.” 

ANSWER: We have amended typos and grammatical and syntactical errors.

Reviewer 2 Report

Major

1.       The quality of the figures is really poor. This needs to be changed before publication is possible. The resolution needs to be much higher, it’s quite difficult to judge the data in the manuscript as is now.

2.       Please indicate for each figure, where applicable, how many independent experiments the data is from/is representative of.

3.       The main figure in this manuscript is figure 3. In figure 3, authors try to show that DRD3 deficiency results in lower IL-2 production by OT-I T cells. However, in my experience, and also in published literature (i.e. 10.1073/pnas.1704227114), OT-I T cells that have been previously stimulated and then are re-stimulated with such a high amount of SIINFEKL peptide should produce a significant amount of IL-2. However, the authors of this manuscript only observe a minor shift and attribute the differences they find in anti-tumor capacity due to reduced IL-2 production by TILs. I don’t really agree with this line of reasoning, as the flow cytometry plots in Figure 3 are not at all convincing. Authors need to explain this discrepancy with published literature.

Minor

1.       Line 95 – authors keep referring to their finding as “genetic evidence” which is very confusing to me. No gene analysis/RNAseq has been performed. Please consider revising.

2.       Line 354 – this manuscript does not contribute to knowledge regarding dopaminergic signalling regulating tumor growth, but rather dopaminergic signalling regulating T cell effector function, which in turn can impact tumor growth. Please amend this sentence as the way it’s written now is incorrect.

3.       Line 360 – 375 – this part of the discussion seems more introductionary. Authors might consider removing it here/merging this into the introduction, as some of the data is already discussed there as well.

4.       Discussion – there is a large section of the discussion (line 382-410) now dedicated to other dopaminergic receptors on other cell types. I don’t see the relevance here, it rather clutters up the discussion, and would ask the authors to shorten or remove these sections to provide a cleaner discussion.

Author Response

Reviewer 2:

Comments and Suggestions for Authors

Major

  1. The quality of the figures is really poor. This needs to be changed before publication is possible. The resolution needs to be much higher, it’s quite difficult to judge the data in the manuscript as is now.

ANSWER: The resolution of figures has been improved in the new version of the paper.

  1. Please indicate for each figure, where applicable, how many independent experiments the data is from/is representative of.

ANSWER: This information has been added to all the figure legends in the new version of the paper.

  1. The main figure in this manuscript is figure 3. In figure 3, authors try to show that DRD3 deficiency results in lower IL-2 production by OT-I T cells. However, in my experience, and also in published literature (i.e. 10.1073/pnas.1704227114), OT-I T cells that have been previously stimulated and then are re-stimulated with such a high amount of SIINFEKL peptide should produce a significant amount of IL-2. However, the authors of this manuscript only observe a minor shift and attribute the differences they find in anti-tumor capacity due to reduced IL-2 production by TILs. I don’t really agree with this line of reasoning, as the flow cytometry plots in Figure 3 are not at all convincing. Authors need to explain this discrepancy with published literature. 

ANSWER: We thank the reviewer for this discussion. Is true that we observed just a small shift of IL-2 associated MFI in flow cytometry analysis when compared the expression of this cytokine in SIINFEKL-treated Drd3+/+ and Drd3-/- OT-I CD8+ T-cells. A similar shift in the MFI associated with IL-2 immunostaining was observed in flow cytometry analysis of  OT-I CD8+ T-cells treated with 100 nM (Figure 5C from Salerno et al., 2017; 10.1073/pnas.1704227114). However, the histograms shown in figure 3A in our study shows how the whole CD8+ T-cell population is IL-2+  immunostained when treated with 1 ug/mL SIINFEKL (approximately 1 uM), while only a fraction of the CD8+ T-cell population is IL-2+ immunostained when CD8+ T-cell are stimulated with 100 nM  SIINFEKL (Figure 5C from Salerno et al., 2017; 10.1073/pnas.1704227114).

Even when there is a small shift in the MFI associated with IL-2 in Drd3-/- CD8+ T-cells compared to Drd3+/+ CD8+ T-cells, our data indicates that Drd3-deficiency in CD8+ T-cells results in a significant decrease in the expression of CD25 and IFN-gamma, and these effects are abrogated when exogenous IL-2 is added to the cell culture. Moreover, to reinforce the evidence indicating that Drd3-deficiency results in impaired production of IL-2 by CD8+ T-cells, we performed new experiments in which Drd3+/+ and Drd3-/- CD8+ T-cells were in vitro activated and the extent of IL-2 secretion was determined in the supernatant by ELISA. The results show that Drd3-deficiency results in a substantial decrease in the secretion of IL-2. These new experiments, which confirm our results obtained by flow cytometry analysis of IL-2, are included in the figure 3B in the new version of the paper.

Minor

  1. Line 95 – authors keep referring to their finding as “genetic evidence” which is very confusing to me. No gene analysis/RNAseq has been performed. Please consider revising.

ANSWER: We have revised the cited sentence to improve the clarity: “Our findings provide evidence indicating that the genetic deficiency of DRD3 impairs the production of IL-2 by CD8+ T-cells, which was associated with attenuated expansion and acquisition of effector function by these cells, and with reduced efficacy of the anti-tumour response in a melanoma mouse model.”

  1. Line 354 – this manuscript does not contribute to knowledge regarding dopaminergic signalling regulating tumor growth, but rather dopaminergic signalling regulating T cell effector function, which in turn can impact tumor growth. Please amend this sentence as the way it’s written now is incorrect.

ANSWER: According to the reviewer’s recommendation, the indicated sentence has been amended: “Thus, these findings contribute to the basic knowledge of how dopaminergic signalling regulates the T-cell response against melanoma and also represent an interesting opportunity for melanoma therapy”.

  1. Line 360 – 375 – this part of the discussion seems more introductionary. Authors might consider removing it here/merging this into the introduction, as some of the data is already discussed there as well.

ANSWER: The indicated paragraph was moved out from the discussion and merged with the second paragraph of the introduction.

  1. Discussion – there is a large section of the discussion (line 382-410) now dedicated to other dopaminergic receptors on other cell types. I don’t see the relevance here, it rather clutters up the discussion, and would ask the authors to shorten or remove these sections to provide a cleaner discussion.

ANSWER: Please note that the reviewer 1 (commentaries 1 and 4) asked us to extend the discussion about dopaminergic regulation through other receptors in inflammation and cancer. For these reasons, we had to keep this section (lines 382-410) and even increase the size of the text dedicated to this issue in the new version of the paper.

Reviewer 3 Report

The manuscript by Chovar et al. deals with dopaminergic signalling in T lymfocytes. The manuscript is interesting and well written. I have following points:

Major:

1) The usage of abbreviations. Every term should be explained in the first appearance. This is the case of abstract (Drd3 deficiency - there is no explication what deficiency it is ) and otherwise in the text (OVA) etc.

2) The volume of tumor. If we assume that this is one half of ellipsoid (three diameters are measrable),  than it is not possible to use formula:  V= (D . d2)/2, but V = (4/3 * π * a * b * c)/2. Please recalculate and change appropriate Figure

3) Although D3 receptor has higher affinity to dopamine, there is very little difference to D4 and D5 receptors (see Myslivecek, Life 2022, 12, 5). Please change the text appropriately.

Minor:

1) Methods: D3 Ko mice were made on which genetical background and how many backcrosses were made ?

Author Response

Reviewer 3:

Comments and Suggestions for Authors

The manuscript by Chovar et al. deals with dopaminergic signalling in T lymfocytes. The manuscript is interesting and well written. I have following points:

Major:

  • The usage of abbreviations. Every term should be explained in the first appearance. This is the case of abstract (Drd3 deficiency - there is no explication what deficiency it is ) and otherwise in the text (OVA) etc.

ANSWER: Drd3 deficiency corresponds to the lack of the gene encoding DRD3. This explanation has been included in the abstract. OVA means chicken ovalbumin, which has also been explained in the text of the new version of the paper. Other abbreviations (previously undefined) were also defined in the text of the new version of the paper, including interleukin 2 (IL-2), interferon g (IFN-g), tumour necrosis factor a (TNF-a), T-cell receptor (TCR).

  • The volume of tumor. If we assume that this is one half of ellipsoid (three diameters are measrable),  than it is not possible to use formula:  V= (D . d2)/2, but V = (4/3 * π * a * b * c)/2. Please recalculate and change appropriate Figure.

ANSWER: It is possible to find different ways to determine the tumour size in the literature using the animal model of melanoma B16, including not only the half ellipsoid (V = (4/3 * π * a * b * c)/2), but also  V= (D . d2)/2 (Enamorado et al., 2016; Park et al., 2019). Some authors even use only the largest diameter (Malik et al., 2017). We cited the indicated references (Enamorado et al., 2016; Park et al., 2019) in the section 2.7 in the new version of the paper, where we described how the tumour size was calculated.

References for this point:

Park, S. L. et al. Tissue-resident memory CD8(+) T cells promote melanoma-immune equilibrium in skin. Nature. 565 (7739), 366-371, doi:10.1038/s41586-018-0812-9, (2019).

Enamorado, M. et al. Enhanced anti-tumour immunity requires the interplay between resident and circulating memory CD8(+) T cells. Nat Commun. 8 16073, doi:10.1038/ncomms16073, (2017).

Malik, B. T. et al. Resident memory T cells in the skin mediate durable immunity to melanoma. Sci Immunol. 2 (10), doi:10.1126/sciimmunol.aam6346, (2017).

  • Although D3 receptor has higher affinity to dopamine, there is very little difference to D4 and D5 receptors (see Myslivecek, Life 2022, 12, 5). Please change the text appropriately.

ANSWER: This has been commented (and the reference cited; see reference 4) in the introduction and discussion of the new version of the paper.

Minor:

  • Methods: D3 Ko mice were made on which genetical background and how many backcrosses were made ?

ANSWER: Drd3-/- mice were initially generated in the 129SvJ strain (Joseph et al., 2002), and then backcrossed in the C57BL/6 genetic background for more than ten generations. This information has been included in the section 2.1. in the new version of the paper.

References for this point.

Joseph JD, Wang YM, Miles PR, Budygin EA, Picetti R, Gainetdinov RR, et al. Dopamine autoreceptor regulation of release and uptake in mouse brain slices in the absence of D(3) receptors. Neuroscience. 2002;112:39-49.

Reviewer 4 Report

The manuscript by Chovar et al. explores the role of DRD3 (dopamine receptor) in CD8 T cells using a single model approach. The manuscript is well-constructed, the introduction and discussion are informative and the results novel. However, certain weaknesses should be addressed to support the authors' conclusions.

Major points:

1- Single antigen/single tumor. Relying entirely on the avid reactivity of the OT-1 TCR casts doubts on whether the authors' findings can be generalized. Figure 1 and 3 show relatively modest differences with such strong antigenic stimulation. Given the mice accessible to the investigators, adding CD8 T-cell responses towards other antigens (or even towards SIINFEKL peptide variants recognized with less avidity) should be feasible. This is essential to alleviate this reviewer's concerns about the overall relevance of the findings.   

2- The evoked IL-2 centered mechanism needs clarifications. As presented, the data do not support the conclusions (written in the title of the manuscript). Both increased CD25 and IFNg expression could be the consequence of increased T-cell activation (which is not assessed neither phenotypically, biochemically nor using gene expression) and the correction of the phenotype by IL-2 does not prove that auto/paracrine secretion is responsible for higher IFNg production in physiological settings (as these tests were performed in vitro with no clear idea about the concentration of DRD3 ligands present in culture relative to in vivo). The authors should either modify their conclusions and title or provide more evidence linking the presence of DRD3 to IL-2 expression leading to anti-tumor immunity (only an association can be described at this time).     

3- Figure 4. Results suggest that 4 (or perhaps 5) mice per group were used. How many times was the experiment done? A single experiment with 4-5 mice per group would be insufficient to conclude. Results from two or three experiments with 4-5 mice per group each time would be a minimal requirement.  

Minor points: 

1- Several sentences were hard to understand. Editing is recommended

2- Methods. It seems clear from the results, but the authors should specify that their OT-1 mice are not on B6-Rag background. 

3- References. Careful revisions are required. For example, Ref. 20 is published, not "accepted". 

Author Response

REVIEWER 4

Comments and Suggestions for Authors

The manuscript by Chovar et al. explores the role of DRD3 (dopamine receptor) in CD8 T cells using a single model approach. The manuscript is well-constructed, the introduction and discussion are informative and the results novel. However, certain weaknesses should be addressed to support the authors' conclusions.

Major points:

  • Single antigen/single tumor. Relying entirely on the avid reactivity of the OT-1 TCR casts doubts on whether the authors' findings can be generalized. Figure 1 and 3 show relatively modest differences with such strong antigenic stimulation. Given the mice accessible to the investigators, adding CD8 T-cell responses towards other antigens (or even towards SIINFEKL peptide variants recognized with less avidity) should be feasible. This is essential to alleviate this reviewer's concerns about the overall relevance of the findings.   

ANSWER: We have modified our conclusions in the title, abstract, last paragraph of the introduction, and first paragraph of the discussion. The conclusions of our findings have been changed from a broader scope in cancer and anti-tumour immunology to a scope limited to a mouse melanoma model. 

  • The evoked IL-2 centered mechanism needs clarifications. As presented, the data do not support the conclusions (written in the title of the manuscript). Both increased CD25 and IFNg expression could be the consequence of increased T-cell activation (which is not assessed neither phenotypically, biochemically nor using gene expression) and the correction of the phenotype by IL-2 does not prove that auto/paracrine secretion is responsible for higher IFNg production in physiological settings (as these tests were performed in vitro with no clear idea about the concentration of DRD3 ligands present in culture relative to in vivo). The authors should either modify their conclusions and title or provide more evidence linking the presence of DRD3 to IL-2 expression leading to anti-tumor immunity (only an association can be described at this time). 

ANSWER: The title and conclusions (in the abstract, last paragraph of the introduction, and first paragraph of discussion) have been changed indicating that DRD3-signalling is associated with higher IL-2 production and a more potent anti-tumour response mediated by CD8+ T-cells in a mouse model of melanoma.

  • Figure 4. Results suggest that 4 (or perhaps 5) mice per group were used. How many times was the experiment done? A single experiment with 4-5 mice per group would be insufficient to conclude. Results from two or three experiments with 4-5 mice per group each time would be a minimal requirement.  

ANSWER: Experiments from figure 4 (figure 5 in the new version of the paper) were performed twice, using 5-7 mice per group each time. These results are currently shown separated by sex, including the results obtained from males in the figure 5 and females in the figure S4 in the new version of the paper. Of note, similar results were obtained in both genders.

Minor points: 

  • Several sentences were hard to understand. Editing is recommended

ANSWER: we have improved the grammar throughout the whole manuscript.

  • It seems clear from the results, but the authors should specify that their OT-1 mice are not on B6-Rag background. 

ANSWER: In the section 2.1. we have stated that “Of note, OT-I mice (strain 003831 from Jackson Laboratories), which harbour chicken ovalbumin (OVA)-specific CD8+ T-cells expressing a transgenic T-cell receptor (TCR) bearing the variable chains Va2 and Vb5 with specificity for the MHC/peptide complex H-2Kb/OVA257-264, were not developed in the Rag1-/- background.”

  • Careful revisions are required. For example, Ref. 20 is published, not "accepted".

ANSWER: The references have been carefully revised.

Round 2

Reviewer 2 Report

No further comments - I stand by my earlier comment that the discussion is a bit warped now, but the data presented, with the addition ELISA, are now convincing.

Reviewer 4 Report

My concerns are largely alleviated as the conclusions now better reflect the data.